# The Range and Direction of Changes in the Classification of the Body Mass Index in Children Measured Between the Ages of 6 and 10 in Gdansk, Poland (Longitudinal Studies)

**DOI:** 10.3390/nu17152399

**Published:** 2025-07-23

**Authors:** Marek Jankowski, Aleksandra Niedzielska, Jacek Sein Anand, Beata Wolska, Paulina Metelska

**Affiliations:** 1Health Promotion and Addiction Prevention Center, 80-409 Gdansk, Poland; marek.jankowski@opz.gdansk.pl (M.J.); a.niedzielska@opz.gdansk.pl (A.N.); 2Division of Clinical Toxicology, Faculty of Health Sciences with the Institute of Maritime and Tropical Medicine, Medical University of Gdansk, 80-211 Gdansk, Poland; jacek.sein_anand@gumed.edu.pl; 3Department of Martial Art, Gdansk University of Physical Education and Sport, 80-336 Gdansk, Poland; beata.wolska@awf.gda.pl; 4Department of Public Health & Social Medicine, Faculty of Health Sciences with the Institute of Maritime and Tropical Medicine, Medical University of Gdansk, 80-211 Gdansk, Poland

**Keywords:** anthropometry, childhood obesity, nutritional assessment, development

## Abstract

**Background/Objectives:** Body Mass Index (BMI) is a widely used indicator of children’s nutritional status and helps identify risks of being underweight and overweight during development. Understanding how BMI classifications evolve over time is crucial for early intervention and public health planning. This study aimed to determine the scope and direction of changes in BMI classification among children between the ages of 6 and 10. **Methods:** This longitudinal study included 1026 children (497 boys and 529 girls) from Gdansk, Poland. Standardized anthropometric measurements were collected at ages 6 and 10. BMI was calculated and classified using international reference systems (IOTF and OLAF). BMI classification changes were analyzed using rank transformations and Pearson correlation coefficients (*p* < 0.05) to explore relationships between body measurements. **Results:** Most children (76.51%) retained their BMI classifications over the four-year period. However, 23.49% experienced changes, with boys more often moving to a higher BMI category (15.29%) and girls more frequently shifting to a lower category (14.03%). The prevalence of children classified as living with obesity declined between ages 6 and 10, while both overweight and underweight classifications slightly increased. Strong correlations were observed between somatic features and BMI at both ages. **Conclusions:** The stability of BMI classification over time underscores the importance of early identification and sustained monitoring of nutritional status. The sex-specific patterns observed highlight the importance of targeted health promotion strategies. In this context, incorporating dietary interventions—such as promoting balanced meals and reducing unhealthy food intake—could play a significant role in maintaining healthy BMI trajectories and preventing both obesity and undernutrition during childhood.

## 1. Introduction

Height and body mass in the period of growth and development indicate health and nutrition of an individual, as well as reflect the overall welfare of a population [1,2]. The Quetelet II index—commonly known as the Body Mass Index (BMI)—is defined as body mass in kilograms divided by the square of height in meters. It is the most frequently cited index in the literature and most used in practice for the assessment of biological development [3,4]. The Body Mass Index is considered a good indicator of underweight, overweight, or obesity in children and adolescents, provided that reference systems appropriate to the developmental period of the child are applied [3,5]. A child’s body weight in relation to the square of their height is a useful tool that accurately reflects their nutrition [5,6,7], which can be one of the indicators of the child’s health potential. The relationship between BMI and precise methods of measuring body fat in children appears to be well documented [4,8]. Among the various methods of indirect assessment of body fat, especially in epidemiological studies, both in adults and children, BMI is the most frequently used one [2,9,10,11]. The usefulness of BMI in predicting the risk of obesity in adulthood has been confirmed [12,13]. This index is recommended as a tool to assess body mass by the WHO, the CDC, the IOTF, and the European Childhood Obesity Group (ECOG) [9,14]. It is important to keep in mind, however, that BMI is more of an indication of increased body mass rather than excess adipose tissue relative to height [4,15]. Monitoring BMI changes in children, combined with appropriate behavioral interventions, forms the basis for preventing eating and weight-related disorders [13,16,17].

Data on somatic features enable the observation of secular trends and consider risk factors for the development of lifestyle diseases. Such factors include excess body weight, the global incidence of which has been rapidly increasing in recent years [14,18,19,20]. Research shows that one in four children in Europe are overweight [5,21]. The report from the International Obesity Task Force (IOTF), published in 2004, containing data from 22 European countries, indicates that, in some countries, the percentage of overweight children aged 7–10 years accounts for over 30% of the population [14,22]. Childhood overweight and obesity, influenced by a complex interplay of factors including social, environmental, genetic, and behavioral elements, represent a significant public health challenge. Numerous studies confirm that children living with excess body mass are at a higher risk of many negative health outcomes in the future [10,13,23,24]. Monitoring somatic features (height and body mass) allows for not only the early diagnosis of disorders related to excess body mass, but also pathological conditions related to its deficiency. The reasons for body mass deficiency in children are usually multifaceted. They are most often connected to malabsorption, endocrine diseases, incorrect eating behaviors, poverty, and disorders related to emotional tension [5,25]. It is noteworthy that studies on the prevalence of overweight and obesity in children far outnumber those addressing underweight or malnutrition [2,26,27,28].

In addition to its role as a marker of somatic development, BMI reflects the cumulative effects of nutrition, physical activity, and metabolism. Nutritional factors—such as energy intake, dietary quality, and eating behaviors—can significantly influence BMI trajectories in children. Therefore, including a discussion of diet-related determinants is essential for a comprehensive understanding of BMI changes during childhood [2,13,17,29].

Recent studies underscore that shifts in BMI among children may be driven not only by changes in weight-to-height ratios but also by dynamic developments in neuromotor coordination, metabolic adaptation, and diet quality [30]. Additionally, dietary composition—especially macronutrient balance—has been shown to modulate anthropometric trajectories beyond simple caloric intake [31]. Finally, Monda et al. observed that resting energy expenditure remains preserved relative to fat-free mass in individuals with severe obesity, indicating metabolic adaptations beyond BMI alone [32].

### Aim of the Study

We aim to determine the scope and direction of changes in the Body Mass Index (BMI) classification in children examined at the ages of 6 and 10.

## 2. Materials and Methods

This study was a longitudinal observational study conducted in Gdansk, Poland, over two periods: 1999–2005 and 2003–2008. The data were collected as part of local public health monitoring programs implemented by the Center for the Promotion of Children’s Health and Fitness.

A total of 1026 children (497 boys and 529 girls) were examined twice—first at age 6 (range 6.00–6.99 years) and again at age 10 (range 10.00–10.99 years). Children with any condition that impeded accurate anthropometric measurements (e.g., limb casts, musculoskeletal disorders) were excluded.

Body height was measured using a stadiometer accurate to 1 mm in the Frankfurt plane position. Body weight was measured to the nearest 50 g using Polish-made TYP WB 150 and Mensor TYP WE 150 digital medical scales, both equipped with stadiometers. Devices were calibrated daily.

All measurements were taken at the Center under standardized conditions by trained staff. Written informed consent was obtained from parents or legal guardians, and all procedures were conducted in their presence.

The Center is a local government unit. It offers the following services: pediatric examinations (including assessment of the child’s body posture), anthropometric measurements, nutritional analysis, psychological assessment, and assessment of physical fitness and physical capacity.

The Center is authorized to conduct all of the above-mentioned examinations as part of its statutory activities. Written consent from the child’s parent or guardian was required, and all examinations were conducted in the presence of the parent or guardian.

The size of the study group divided by gender is shown in Table 1.

International developmental norms for excess body mass [3] and body mass deficiency for children and adolescents aged 2–18 years [33] were used as a frame of reference for the Body Mass Index (BMI).

Percentile values for height, weight, and BMI for the examined persons were determined based on current percentile grids for the population of children and adolescents in Poland—the OLAF program [34].

Due to the lack of percentile values for the age range between 6.00 and 6.49 in the OLAF percentile grids (OLAF percentile grids apply to people aged 6.5–18.5), the data from 6-year-old children in the range of 6.00–6.49 were not used for statistical analysis using percentile values. Although these children were included in raw BMI comparisons, they were excluded from percentile-based OLAF analysis due to the age range mismatch. Their exclusion had minimal effects on the overall trend direction but may affect precision near category boundaries.

The basic analysis of somatic characteristics data was carried out separately for boys and girls, taking into account groups distinguished based on the Body Mass Index (BMI) classification. The following basic characteristics of the distributions of measurable features were calculated: arithmetic mean, median, standard deviation, and minimum and maximum values.

Two transformations were used to analyze the dynamics of change: ranking—scale change transformation to a weaker one which enables the presentation of selected regularities of the phenomenon; differentiation—the transformation which quantitatively expresses changes between the group of the same children at the ages of 6 and 10, so on the basis of arithmetic sequence differences, it was possible to perform numerical calculations, including distribution characteristics and correlation analysis.

The relationship between somatic features and the Body Mass Index (BMI) was examined based on the Pearson correlation coefficient. It was assumed that the correlation coefficients were statistically significant at the level of *p* < 0.05. Basic statistical analyses were performed for the percentile values of anthropometric measurements and the BMI for individual BMI classification groups and their mutual linear correlation.

In order to observe the number, percentage, and direction of changes in BMI classification in the same children examined at two life stages, ranks for BMI (body mass classification) were determined. The rank assignment is shown in Table 2.

Observing the rank difference results for BMI enables the analysis of the scope of changes in body mass classification and their direction. The observed changes towards positive values indicate a classification change connected to relative (in reference to the IOTF reference system) body mass increase for age and gender. Changes towards negative values indicate a change in body mass classification associated with a relative decrease in body mass for gender, age, and height.

Example: The BMI of a child examined at the age of 6 was classified as ‘normal body mass’ (BMI1 = rank 4), but the BMI of the same child at the age of 10 was classified as ‘obesity’ (BMI2 = rank 6). BMI2 − BMI1 (6 − 4 = 2) means a change in body mass classification by two ranks in the direction of relative body mass increase.

Justification for the choice of statistical method: The rank transformation and analysis of directional changes in BMI classification were chosen for data analysis, considering that the method was appropriate for the nature of the data. Given that the BMI categories are ordinal and the main objective of the study was to examine the direction, frequency, and magnitude of changes observed in individual children over time, a descriptive and rank-based method was chosen rather than inferential modeling. Furthermore, due to the structure of the data set—which includes categorical BMI classifications at two time points rather than continuous repeated measures—the use of ANOVA with repeated measures or its nonparametric alternatives was rejected because it would not accurately reflect the main dynamics of change that were being studied. The applied method therefore allows for a more accurate reflection of actual changes in BMI classification at the individual level, which is particularly important in the context of assessing the effectiveness of intervention activities or monitoring health trends in the child population.

## 3. Results

A lower percentage of people with normal body weight was observed in children aged 10 (69.88%) compared to studies conducted in the same population at the age of 6 (71.23%).

At 10 years of age, the percentage of children with excess (17.45% at the age of 6 vs. 18.51% at the age of 10) and deficient (8.97% at the age of 6 vs. 11.59% at the age of 10) body mass increased compared to the results achieved at the age of 6.

The change in the percentage of individual body weight classifications varied in direction and scope depending on gender.

At the age of 10, a lower percentage of boys and girls with obesity was observed, while the percentage of overweight children increased. The highest percentage (4.73%) of children with obesity was found in the group of girls aged 6; however, the percentage of obesity in the same group of girls at the age of 10 was only 1.70%. A lower incidence of obesity at the age of 10 was also observed in boys. Compared to the percentage of boys with this body weight classification at the age of 6, when the percentage of boys with obesity was 3.62%, at the age of 10, it dropped to 2.62%.

In boys, the number and percentage of children classified as living with underweight slightly decreased (from 9.25% at the age of 6 to 8.25% at the age of 10). However, in girls, the percentage of people with body weight deficiency increased from 8.97% at the age of 6 to 11.59% at the age of 10. The number and percentage of children in each body mass classification rank are presented in Table 3.

The majority of 10-year-old children (76.51%) maintained their body mass classification from when they were 6. This observation was almost equal between boys (75.45%) and girls (76.51%). One fourth of the children participating in the study changed their body mass classification, and the direction of this change turned out to be different in boys and girls. The main direction of changes in body mass classification in boys was achieving a higher level of BMI classification (i.e., relative weight gain). It concerned 15.29% of the participants. In girls (14.03%), the main direction of changes turned out to be the opposite and concerned achieving a lower level of BMI classification. Changes in body mass classification were primarily associated with a change of one level. Only 14 children, i.e., 1.36%, changed their body mass classification by two levels. Change in body mass classification by three levels concerned only two girls. The detailed data on changes in body mass classification can be found in Table 4.

In order to expand on the observations of changes in somatic features and BMI, a statistical analysis of percentile values was performed. The results of the analysis are presented in Table 5 and Table 6. The mean value of the body height percentile for boys aged 6 (6.50–6.99) was 44.2. After 3 years, the same group of boys had a higher mean body height percentile value of 52.1. A higher mean percentile value in boys at the age of 10 can also be observed in body mass. The mean BMI percentile, however, was lower in boys at the age of 10 than at the age of 6, which resulted from the difference in the increased proportion between the mean percentile values of body mass and height. In the group of girls, similarly to the population of boys, the mean BMI percentile was lower at the age of 10, decreasing from 55.8 at the age of 6 to 51.0. Similarly, a lower mean body mass percentile was observed in girls at the age of 10 compared to the age of 6. Of the somatic features examined in this group, only the mean value of the body height percentile was higher at the age of 10. The mean percentile values of the BMI in boys with obesity, overweight, or normal weight at the age of 6 were lower at the age of 10, while in boys who were slim or significantly underweight, the mean percentile values were higher after 3 years of observation. Similarly to the group of boys, the mean BMI percentile values in girls with obesity, overweight, and normal body weight at the age of 6 turned out to be lower after 3 years of observation. It has also been observed that children (boys and girls) with excess body mass (overweight, obesity) had the highest mean body height values at both 6 and 10 years of age.

The BMI and the somatic features (height and body mass) of boys and girls examined in two periods of life—at 6 and 10 years of age—are statistically significantly correlated. Detailed data are provided in Table 7 and Table 8.

## 4. Discussion

The presented research shows that most children (over 75%) aged 10 maintain their BMI classification assigned at the age of 6. Only a quarter of them changed their BMI classification. For most boys, this change was associated with a relative (to age and body height) weight gain, while for girls, the direction of change turned out to be the opposite. For both sexes, it was found that the percentage of children with obesity was lower in 10-year-olds. Although this observation only pertains to the prepubescent period, it appears to contrast with some studies suggesting that obesity prevalence increases with age [35]. This may be related to the so-called growth spurt that starts around the age of 10. A rapid increase in body height may result in temporarily lower values of the BMI and a change in the BMI classification from obesity to excess body mass, although existing reference systems created on the basis of population studies (including the one used in this work) largely take into account the existence of this phenomenon. The observation of the same children over a longer period will enable the verification of this hypothesis. According to many authors, obesity is very common in adults who were overweight in childhood [10,24,36,37]. It is worth noting that only 6% of children who were classified as living with overweight or obesity at the age of 5 had a normal body weight at the age of 14 [38], but there was a tendency for girls after puberty to obtain lower classification of BMI values in the long term (from 7 to 18 years of age) [39].

Previous research shows that somatic features are genetically determined, but the strength of this conditioning is different for height and body mass. Strong conditioning is found in the case of bone dimensions, while weak conditioning is found in the case of body mass [40]. The biological development of children in terms of a significantly determined “developmental pathway” for height and body mass is confirmed by the research results presented in this work: there is a high linear correlation of measurement values and percentile values for height, body mass, and BMI at the ages of 6 and 10.

Although BMI has limitations, it remains one of the most frequently used indicators in population health research due to its simplicity, cost-effectiveness, and ease of use in international comparisons [2,4]. It is widely applied in large-scale health programs, allowing rapid identification of children who may require further diagnosis. Nevertheless, BMI alone does not capture the complexity of cardiovascular risk in children. Recent research shows that this risk is also associated with waist circumference, waist-to-height ratio, blood pressure, lipid profile, and insulin resistance [26,41]. These indicators may more accurately predict future cardiovascular risk than BMI alone.

In both clinical practice and public health, a comprehensive approach to monitoring—including anthropometric, metabolic, and lifestyle assessments—is increasingly promoted [13,27,28].

Only this approach allows for precise preventive and therapeutic interventions. From a population-level perspective, BMI’s simplicity and low cost continue to make it an invaluable screening tool. However, for children with BMI values outside the normative range, further assessment of metabolic and cardiovascular risk factors is warranted.

However, motor adaptability and perceptual–cognitive feedback mechanisms—such as those discussed in Mancini et al. (2024) in the context of sensorimotor training—may also influence developmental trajectories and should be considered when evaluating BMI classification shifts in children [42].

It is important to recognize the potential impact of dietary interventions on BMI development in children. Numerous studies have shown that improvements in diet quality—including increased consumption of fruits and vegetables, reduction in sugar-sweetened beverages, and promotion of regular meal patterns—can positively affect weight status [5,43]. Furthermore, school-based nutrition programs and family-centered dietary guidance have demonstrated effectiveness in slowing BMI gain or promoting weight normalization in children [13,44,45]. Considering that almost one-quarter of the children in our study experienced a shift in BMI classification, early nutritional interventions could play a critical role in stabilizing or reversing unhealthy trends, particularly in children with upward or downward shifts in BMI classification.

This study has several limitations that should be acknowledged to provide context for the findings:The sample was drawn exclusively from Gdansk, Poland, which may limit the generalizability of the results to broader populations or different cultural and geographic contexts.This study was limited to two data collection points (ages 6 and 10), which restricts the ability to assess BMI development trajectories across different stages of childhood.BMI classification systems (IOTF and OLAF) used in this study, while internationally recognized, may not fully reflect body composition (e.g., fat distribution or muscle mass), potentially limiting the precision of nutritional status assessment.The absence of data on lifestyle factors such as physical activity levels, dietary patterns, and socioeconomic status prevents a more comprehensive understanding of the determinants of BMI changes.The statistical approach used focused on descriptive analysis and rank-based transformations. While appropriate for categorical data, it does not allow modeling causal relationships or detecting interactions between variables.Although the data used in this study were collected several years ago, it is important to note that the study was conducted in a longitudinal design, with each child assessed at two different time points. This allowed us to capture within-individual changes in nutritional status, including transitions between BMI categories. With this method of data collection, the presence and direction of changes, as well as the observed trends, are not dependent on the currency of the growth reference charts. Nevertheless, potential differences between past and current anthropometric conditions in the population should be acknowledged as a limitation when interpreting the applicability of the findings to the present context.

## 5. Conclusions

The observed stability of BMI classification in most children between ages 6 and 10 highlights the importance of early identification of both excess and deficient body mass in the youngest age groups possible, and to undertake health-promoting interventions for them aimed at developing their ability to maintain appropriate body weight.

The change in the classification of body mass in some children with obesity at the age of 6 to overweight at the age of 10 (even if it is associated with a periodic, significant increase in body height at this age) may offer a window of opportunity for implementing health interventions aimed at obesity prevention.

Integrating nutritional strategies into public health interventions—including education on healthy eating habits and access to balanced meals—may further enhance efforts to prevent or reduce childhood overweight and obesity, especially during key developmental windows such as early and middle childhood.

Throughout this study, person-first and non-stigmatizing language is used to respect the dignity and individuality of all children, in line with international guidance on addressing weight-related bias.

## Figures and Tables

**Table 1 nutrients-17-02399-t001:** Numerical characteristics of the study group.

(*n*)		(%)		Sum
Boys	Girls	Boys	Girls	
497	529	48.44	51.56	1026

**Table 2 nutrients-17-02399-t002:** Rank assignment for individual BMI classification.

BMI Classification [2,22]	Rank
Obesity	6
Overweight	5
Normal weight	4
Thinness	3
Underweight	2
Significantly underweight	1

**Table 3 nutrients-17-02399-t003:** Number and percentage of children ranked according to body mass classification performed at the ages of 6 and 10.

	Boys				Girls				Sum (Boys and Girls)			
	Aged 6		Aged 10		Aged 6		Aged 10		Aged 6		Aged 10	
	(*n*)	(%)	(*n*)	(%)	(*n*)	(%)	(*n*)	(%)	(*n*)	(%)	(*n*)	(%)
body mass classification												
obesity	18	3.62	13	2.62	25	4.73	9	1.7	43	4.19	22	2.14
overweight	65	13.08	89	17.91	71	13.42	79	14.93	136	13.25	168	16.37
normal weight	368	74.05	354	71.23	387	73.16	363	68.63	755	73.59	717	69.88
thinness	41	8.25	35	7.04	39	7.37	63	11.91	80	7.8	98	9.56
underweight	2	0.4	6	1.20	3	0.57	9	1.7	5	0.49	15	1.47
significantly underweight	3	0.6	0	0.00	4	0.75	6	1.13	7	0.68	6	0.58
sum	497	100	497	100	529	100	529	100	1026	100	1026	100

**Table 4 nutrients-17-02399-t004:** Numerical characteristics of the scope and direction of changes in the classification of the BMI in children examined at 6 and 10 years of age.

	Boys		Girls		Sum (Boys and Girls)	
	Number of Rank Changes BMI2 − BMI1	Percentage of Rank Changes BMI2 − BMI1	Number of Rank Changes BMI2 − BMI1	Percentage of Rank Changes BMI2 − BMI1	Number of Rank Changes BMI2 − BMI1	Percentage of Rank Changes BMI2 − BMI1
	(*n*)	(%)	(*n*)	(%)	(*n*)	(%)
range of rank changes						
−3	0	0	1	0.19	1	0.10
−2	2	0.4	4	0.76	6	0.58
−1	51	10.26	86	16.26	137	13.35
0	375	75.45	410	77.50	785	76.51
1	62	12.48	26	4.91	88	8.58
2	7	1.41	1	0.19	8	0.78
3	0	0	1	0.19	1	0.10
sum	497	100	529	100	1026	100

**Table 5 nutrients-17-02399-t005:** Mean percentile values of body mass, body height, and BMI.

	(*n*)	(%)	mean					
			Percentile BMI1	Percentile BMI2	Percentile Body Mass 1	Percentile Body Mass 2	Percentile Body Height 1	Percentile Body Height 2
Boys	270	48	55.2	52.5	51.09	52.2	44.2	52.1
Girls	288	52	55.8	51.0	54.46	52.0	50.9	53.4
sum/average	558	100	55.5	51.75	52.77	52.1	47.55	52.75

**Table 6 nutrients-17-02399-t006:** Mean percentile values of body mass, body height, and BMI depending on BMI classification at the age of 6.

	Mean													
	Boys							Girls						
Classification BMI 1		Percentile BMI 1	Percentile BMI 2	Percentile Body Mass 1	Percentile Body Mass 2	Percentile Body Height 1	Percentile Body Height 2		Percentile BMI 1	Percentile BMI 2	Percentile Body Mass 1	Percentile Body Mass 2	Percentile Body Height 1	Percentile Body Height 2
	(*n*)							(*n*)						
obisity	11	97.9	93.4	96.3	93.9	78.7	79.1	18	98.2	91.8	97.3	91.5	75.9	74.4
overweight	46	89.1	84.4	80.4	78.9	49.9	58.5	40	90.9	83.3	87.4	82.0	63.9	66.7
normal weight	188	50.6	47.3	46.1	47.4	41.8	49.9	203	51.4	46.6	50.0	48.0	49.2	52.2
thinness	23	9.1	15.8	15.4	21.9	34.6	44.8	24	9.8	10.2	10.8	11.4	25.2	27.4
underweight	0	0	0	0	0	0	0	2	1.5	1.1	5.0	3.0	30.5	27.0
significantly underweight	2	0.1	3.5	7.0	13.0	60.0	54.5	1	1.0	0.1	13.0	8.0	76.0	71.0

**Table 7 nutrients-17-02399-t007:** Linear correlation coefficients (r_xy_) of the studied variables in boys (the determined correlation coefficients are significant with *p* < 0.05).

	Body Mass 1	Body Height 1	Body Mass 2	Body Height 2	BMI 1	BMI 2
body mass 1	1.00	0.69	0.79	0.56	0.88	0.62
body height 1	0.69	1.00	0.58	0.80	0.28	0.29
body mass 2	0.80	0.58	1.00	0.60	0.68	0.87
body height 2	0.57	0.80	0.60	1.00	0.24	0.17
BMI 1	0.88	0.28	0.68	0.24	1.00	0.64
BMI 2	0.62	0.29	0.86	0.17	0.64	1.00

**Table 8 nutrients-17-02399-t008:** Linear correlation coefficients (r_xy_) of the studied variables in girls (the determined correlation coefficients are significant with *p* < 0.05).

	Body Mass 1	Body Height 1	Body Mass 2	Body Height 2	BMI 1	BMI 2
body mass 1	1.00	0.72	0.86	0.63	0.89	0.78
body height 1	0.72	1.00	0.62	0.82	0.32	0.35
body mass 2	0.86	0.62	1.00	0.71	0.77	0.92
body height 2	0.63	0.82	0.71	1.00	0.33	0.37
BMI 1	0.89	0.32	0.77	0.33	1.00	0.83
BMI 2	0.78	0.35	0.92	0.37	0.83	1.00

## Data Availability

Data were collected and stored at the Gdańsk Center for Health Promotion and Addiction Prevention. Data are not made public due to patient confidentiality.

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
