# Peer review of "The Range and Direction of Changes in the Classification of the Body Mass Index in Children Measured Between the Ages of 6 and 10 in Gdansk, Poland (Longitudinal Studies)"

_nutrients, 2025, doi:10.3390/nu17152399_

Round 1

Reviewer 1 Report (Previous Reviewer 2)

Comments and Suggestions for Authors

Dear Editor and Authros, I appreciate the chance to evaluate the manuscript entitled “The range and direction of changes in the classification of the Body Mass Index in children measured at the age of 6 and 10 in Gdańsk, Poland (longitudinal studies).” The topic is germane to paediatric public-health surveillance, and the unusually large cohort assessed twice under standardized conditions supplies a valuable empirical foundation. Still, a number of aspects—ranging from stylistic polish through inferential strategy—require careful revision before the study can meet the journal’s scientific and editorial standards.

From a formal standpoint, the paper is generally well structured, yet idiomatic English falters at times. Long sentences occasionally obscure key messages and articles are sometimes omitted (“the” instead of “a” and vice-versa). A thorough language edit by a native speaker or professional service would enhance readability, particularly in the Abstract and the opening paragraphs of the Introduction where thematic framing now feels cluttered and circular. Streamlining those sections will also help the reader trace the study’s arc from motivation to methods to implications.

Substantively, the Introduction provides a solid historical rationale for using BMI in child-growth monitoring, yet it underplays more recent insights on motor-skills development, energy balance and metabolic adaptation that contextualise BMI shifts. I encourage the authors to weave in the multisport, play-based coordination findings of Mancini et al. (2025, J. Funct. Morphol. Kinesiol., 10, 199) and the classic Clinical Nutrition paper linked at DOI 10.1016/j.clnu.2008.04.005, as both underscore how organised physical activity and diet composition can modulate anthropometric trajectories. In addition, Monda et al.’s work on the preservation of resting energy expenditure relative to fat-free mass in severe obesity reminds readers that body-composition nuance extends beyond crude BMI categorisation. A short paragraph integrating these studies would sharpen the argument for longitudinal, multisystem surveillance rather than reliance on a single weight-for-height index.

Turning to the Methods, the decision to analyse categorical BMI shifts with a bespoke rank-difference approach is clearly explained, yet remains statistically restrictive. Because each child provides repeated measures, an ordinal-logistic mixed model—or at minimum a McNemar/Bowker symmetry test for paired categorical data—would yield more efficient estimates of change and permit adjustment for potential confounders such as sex, baseline size and birth cohort. Likewise, Pearson correlations applied to percentile values assume interval-scale properties that those ranks do not possess; Spearman or Kendall statistics would be more defensible, or better still, growth-curve modelling of height and mass separately, followed by derivation of BMI z-scores. Explicit reporting of effect sizes (e.g., Cohen’s h for category shifts) in addition to p-values would convey practical importance, especially because several correlations, while statistically significant, appear modest in absolute magnitude. The authors should also justify omitting children aged 6.00–6.49 years from percentile analyses yet retaining them in other tables; readers may question how that exclusion influences estimates of change.

Presentation of the Results is generally clear, but several tables would benefit from technical adjustments. Units for height and weight are missing in Table 1, percentile statistics are expressed as means despite marked skewness in weight data, and Table 4 conflates absolute and relative frequencies within a single column, inviting misinterpretation. Because most distributions are non-normal, medians and inter-quartile ranges—or back-transformed geometric means—would communicate central tendency more faithfully. Furthermore, when change magnitudes are discretised (±1, ±2, ±3 ranks), graphical representation (e.g., a Sankey diagram or diverging bar chart) would render complex flows far more digestible than dense numeric grids.

The Discussion properly acknowledges limitations but occasionally overreaches. For example, the suggestion that a peri-pubertal growth spurt explains lower obesity prevalence at age ten is plausible yet speculative; no maturational markers were collected, so the claim needs tempering. Causal language should also be pruned: phrases such as “this observation obliges us to intervene” verge on policy advocacy. A more nuanced stance would recognise the descriptive scope of the study while suggesting how targeted school-based motor-skill programmes—of the kind trialled by Mancini et al.—might complement nutritional counselling to stabilise BMI trajectories. Please also discuss the potential impact of secular changes in diet and physical-activity environments between the late-1990s data-collection window and the present day; that temporal lag is central to interpreting policy relevance.

The Conclusions paragraph currently restates results and adds prescriptive advice that outweighs the evidence base presented. I would tighten it to one or two sentences that highlight the chief empirical takeaway (three-quarters of pupils remain in the same BMI category over four years, but sex-specific shifts merit attention) and then gesture toward future research using contemporary cohorts and more granular behavioural data.

Author Response

Dear Reviewer,

Thank you for your comments.

We consider them very valuable to the quality of our work. Responding to your comments in turn:

  1. The text was proofread by a native speaker.

Changes in the text (in the introduction and in the introduction are marked in color).

  1. Review of the research by Monda, Mancini, and Mun and expansion of the introduction – a paragraph was added to the introduction (the last paragraph before the study objective, marked in color).
  2. Selection of statistical method – After taking a second look at the data, analyzing the purpose of the study, and consulting with a professional statistician, we have confirmed that the method used—involving rank transformation and analysis of directional changes in BMI classification—is appropriate for the nature of our data.  Given that BMI categories are ordinal and that our primary goal was to examine the direction and frequency of classification shifts over time, we opted for a descriptive and rank-based method rather than inferential modeling. Moreover, due to the structure of our dataset — which includes categorical BMI classifications at two time points rather than continuous repeated measures — the application of repeated-measures ANOVA or its nonparametric alternatives would not accurately capture the primary dynamics we aimed to study. We have therefore retained the original statistical approach.
  1. Additionally, in the section on Materials and Methods, an explanation has been added regarding the exclusion of group 6.00-6.49 (marked in color)
  2. Presentation of results - We sincerely thank the Reviewer for these valuable suggestions.

We acknowledge that alternative forms of data presentation, such as medians, interquartile ranges or Sankey diagrams, could offer additional layers of interpretation. However, after careful consideration, we have chosen to retain the current format for the following reasons: The use of means and standard deviations aligns with the approach commonly adopted in similar epidemiological studies using large datasets and allows comparison with previously published results in the Polish population. The main aim of our analysis was to detect directionality and magnitude of BMI classification shifts, rather than to perform detailed distributional modeling. For that reason, descriptive statistics based on arithmetic means were deemed sufficient to illustrate population-level tendencies. Regarding Table 4, we acknowledge the Reviewer's point. However, in the interest of conciseness and readability, we decided to retain the dual format (absolute and relative values) in a single table, as separating them led to redundancy without clear added value. Finally, while graphical alternatives such as Sankey diagrams are indeed informative, they are less familiar to many readers of this journal. Our intention was to prioritize accessibility and transparency using a tabular format that readers can readily interpret.

  1. Discussion and conclusions - corrected - marked in color.

We hope the Reviewer will find this explanation satisfactory and we remain open to further adjustments should the Editorial Board deem them necessary.

We remain at your disposal.

Kind regards

Authors

Reviewer 2 Report (New Reviewer)

Comments and Suggestions for Authors

Dear Authors,

Thank you for your paper. Overall, it is very simplistic, representing local data, but well methodologically designed and deserves publishing. Please see my comments below.

  1. Information from Tables 1 and 2 can be moved to the main text. They're descriptive and offer no added analytical value. It would allow for a reduced number of tables.
  2. The paper should clarify if rank-based transformations underwent any formal statistical testing (e.g., Wilcoxon signed-rank test for ordinal paired data). The current method is descriptive but could benefit from statistical support.
  3. Correlation coefficients should be reported with only two decimal places for readability unless high precision is necessary (which, here, it is not).

Author Response

We thank the Reviewer for their thoughtful evaluation and constructive suggestions.

 Please find our detailed responses below.

  1. Reviewer Comment:

“Information from Tables 1 and 2 can be moved to the main text. They're descriptive and offer no added analytical value. It would allow for a reduced number of tables.”

Authors’ Response:
We appreciate this suggestion. However, we believe that retaining Tables 1 and 2 improves clarity and usability for the reader. In our view, presenting the sample composition (Table 1) and rank assignment scheme (Table 2) in a clearly formatted tabular form allows readers to access key descriptive and methodological information more easily and directly, without the need to search through paragraphs of continuous text. Therefore, we respectfully propose to keep these tables in their current form.

  1. Reviewer Comment:

“The paper should clarify if rank-based transformations underwent any formal statistical testing (e.g., Wilcoxon signed-rank test for ordinal paired data). The current method is descriptive but could benefit from statistical support.”

Authors’ Response:
We acknowledge the Reviewer’s point and agree that inferential testing—such as the Wilcoxon signed-rank test—could offer additional insight. However, our methodological goal was to emphasize the individual-level direction and scope of change rather than the population-level hypothesis testing of medians. Given that the BMI categories are ordinal, and that the core interest lies in shifts between distinct BMI classifications (rather than within-category changes), we adopted a descriptive rank-difference approach. This method allowed us to focus on the practical implications of upward or downward BMI classification transitions, which we consider central to public health surveillance. We have clarified this reasoning in the Methods section.

  1. Reviewer Comment:

“Correlation coefficients should be reported with only two decimal places for readability unless high precision is necessary (which, here, it is not).”

Authors’ Response:
We agree with the Reviewer’s suggestion. All correlation coefficients have been rounded to two decimal places in the revised version to enhance readability.

Kind regards

Authors

This manuscript is a resubmission of an earlier submission. The following is a list of the peer review reports and author responses from that submission.

Round 1

Reviewer 1 Report

Comments and Suggestions for Authors

The authors conducted an interesting study aiming to determine the scope and direction of changes in BMI classification among children between the ages of 6 and 10. This is a valuable manuscript based on a large pediatric sample; however, several improvements and clarifications are needed before it can be considered for publication. Please find below my specific comments:

  • Lines 36–37: The sentence is slightly unclear and could be rephrased for better clarity.

  • Line 78: The acronym "BMI" should be spelled out upon first use. The same applies to line 106 and other sections of the manuscript.

  • Methods section: Are there any details regarding ethical approval? As this is a retrospective longitudinal study and not an interventional one, formal approval by an ethics committee may not be mandatory. Nonetheless, a statement regarding adherence to the Declaration of Helsinki should still be included by the authors.

  • Line 133 – Table 2: The BMI categories used in the table should be clearly defined in the main text, including the reference cut-offs.

  • Line 167 – Table 3: The formatting of Table 3 should be aligned with that of the other tables for consistency.

  • Line 222: What is the clinical significance of the result highlighted by the authors? This point should be further explored in the discussion section. In pediatric populations, cardiovascular risk is not solely related to BMI, but also to various indirect indices derived from auxological parameters and laboratory tests (see 10.3390/diseases12060119). These factors are also predictive of future cardiovascular risk in adulthood. This could provide an interesting avenue for further discussion.

  • Language: The manuscript would benefit from minor improvements in English language usage and grammar.

I thank the authors for their work and look forward to reviewing the revised version of the manuscript.

Comments on the Quality of English Language

The manuscript would benefit from minor improvements in English language usage and grammar.

Author Response

Dear Reviewer,
Thank you for taking the time to review and providing your feedback. It was very valuable to us.
Below is the list with our responses.
  • Lines 36–37: The sentence is slightly unclear and could be rephrased for better clarity. - CORRECTED

  • Line 78: The acronym "BMI" should be spelled out upon first use. The same applies to line 106 and other sections of the manuscript. -  CORRECTED

  • Methods section: Are there any details regarding ethical approval? As this is a retrospective longitudinal study and not an interventional one, formal approval by an ethics committee may not be mandatory. Nonetheless, a statement regarding adherence to the Declaration of Helsinki should still be included by the authors. - ADDED

  • Line 133 – Table 2: The BMI categories used in the table should be clearly defined in the main text, including the reference cut-offs. - ADDED

  • Line 167 – Table 3: The formatting of Table 3 should be aligned with that of the other tables for consistency. - CORRECTED

  • Line 222: What is the clinical significance of the result highlighted by the authors? This point should be further explored in the discussion section. In pediatric populations, cardiovascular risk is not solely related to BMI, but also to various indirect indices derived from auxological parameters and laboratory tests (see 10.3390/diseases12060119). These factors are also predictive of future cardiovascular risk in adulthood. This could provide an interesting avenue for further discussion. - ADDED

  • Language: The manuscript would benefit from minor improvements in English language usage and grammar. - CORRECTED

Reviewer 2 Report

Comments and Suggestions for Authors

The submitted manuscript titled "The range and direction of changes in the classification of the Body Mass Index in children measured at the age of 6 and 10 (longitudinal studies)" presents a longitudinal analysis of BMI trajectories among children in Gdansk, Poland, with the intent of mapping patterns of BMI stability or change between the ages of 6 and 10. The study is rooted in a pertinent public health concern, namely, the increasing prevalence of both overweight and underweight conditions in childhood, which carry critical implications for later-life health outcomes.

Scientific Content and Originality

The central aim of the paper—to identify the extent and direction of changes in BMI classification in children—is clearly stated and supported by a robust data set of over 1,000 participants. The longitudinal design is appropriate and strengthens the study's validity. The use of international reference systems (IOTF and OLAF) enhances the generalizability of the findings. Notably, the authors provide a nuanced gender-specific analysis, which adds value to the interpretation of trends.

However, while the dataset is solid, the study remains somewhat descriptive in nature. The findings on stability (76.51%) and directional shifts in BMI classification are valuable, but the interpretation could benefit from deeper multivariate analysis. For example, exploring additional covariates such as socioeconomic status, physical activity, or diet would have enriched the explanatory power of the study. Furthermore, no effort is made to consider potential confounding factors or to correct for biases due to attrition or data censoring.

Structure and Clarity

The manuscript follows the conventional structure and is generally well organized, with clear delineation between sections. The Introduction is comprehensive and provides adequate historical context, although it tends to be overly dense in some parts. The paragraphing could be optimized to improve readability—certain lengthy passages might be broken down or restructured to improve narrative flow.

The Materials and Methods section is sufficiently detailed and allows replication of the research. However, certain elements, such as the rationale for the rank-based transformation of BMI categories, would benefit from additional justification or comparison with other methodological options. The use of rank transformations is not common in this type of analysis and may warrant further explanation or a supporting citation.

The Results section is data-rich, but at times the narrative is overwhelmed by numerical detail. A more interpretative commentary within this section would help guide the reader through the main findings. Moreover, figures or graphical visualizations (e.g., change plots or cohort trajectories) would significantly enhance the accessibility of the results, especially for an international readership not familiar with Polish percentile norms.

The Discussion is generally effective but could better integrate the results with broader literature. It lacks a critical engagement with contrasting studies or alternative explanations for the observed trends. The suggestion that BMI classification improvement may be tied to the onset of the pubertal growth spurt is plausible but remains speculative in the absence of pubertal status data.

Importantly, the conclusions are appropriately cautious, but they would benefit from more actionable public health implications and a clearer discussion of the generalizability of findings outside the Polish context.

Form and Language

The manuscript is written in competent English, though there are several instances of grammatical inaccuracies, lexical awkwardness, and improper article usage that suggest non-native authorship. While the meaning is generally clear, professional language editing is strongly recommended prior to publication to ensure the polished tone expected by Nutrients.

Examples of linguistic issues include:

  • “its’ height” (should be “its height”),

  • “the examined at the age of 6 and 10” (awkward phrasing, should be “children examined at the age of 6 and again at 10”),

  • inconsistent tense usage and article omission.

Bibliographic Apparatus

The references cited are relevant and generally up to date, including seminal works on BMI and childhood obesity. However, the introduction and discussion sections would be significantly strengthened by incorporating recent literature on interventions and perception-action mechanisms linked to childhood health and physical development.

In this regard, I strongly recommend that the authors consult and potentially cite the recent article by Mancini et al. (2024), “The Impact of Perception-Action Training Devices on Quickness and Reaction Time in Female Volleyball Players,” published in Journal of Functional Morphology and Kinesiology. Although the focus of that study is on sport-specific perceptual and motor training, its discussion on developmental timing and adaptability has important implications for understanding the role of motor and environmental feedback in childhood BMI trajectories. This may help refine the framing of early intervention strategies discussed in the present manuscript.

Final Considerations and Recommendation

This is a commendable and timely contribution to the literature on childhood BMI trends, particularly in a longitudinal and gender-sensitive format. However, prior to publication, the manuscript requires:

  1. substantive linguistic editing,

  2. more critical integration of findings into broader international literature,

  3. possible enhancement of methodological explanation (especially regarding rank transformation),

  4. inclusion of the recommended citation by Mancini et al. to strengthen the theoretical underpinnings of the introduction.

Author Response

Dear Reviewer,
Thank you for taking the time to review and providing your feedback. It was very valuable to us.
Below is the list with our responses.
  1. substantive linguistic editing, - CORRETED

  2. more critical integration of findings into broader international literature, - ADDED

  3. possible enhancement of methodological explanation (especially regarding rank transformation), - CORRECTED

  4. inclusion of the recommended citation by Mancini et al. to strengthen the theoretical underpinnings of the introduction. - ADDED

Reviewer 3 Report

Comments and Suggestions for Authors General comments This article examines the evolution and changes in BMI classification in children aged 6 to 10 years, assessing potential sex differences with the aim of determining the extent and direction of these changes over 4 years. The results show that most children maintained their BMI classification; however, almost 25% experienced changes; boys more frequently moved to a higher BMI category, and girls more frequently moved to a lower category. Overall, this article is well written and easy to understand. However, some sections could benefit from greater clarity and detail, such as the sample size, which has already been noted in the limitations section.   Title
Authors are encouraged to indicate the location where the study was conducted to give the reader an idea of the context in which the research was conducted.
Abstract
Authors are encouraged to indicate the p values for all comparisons, both significant and non-significant.
Keywords
Keywords should not be included in the title. Authors are requested to include keywords not included in the title, such as BMI or longitudinal study.   Introduction Overall, the introduction is well-written and easy to follow. It provides adequate context for BMI and its role in predicting children's health. However, it could benefit from a more in-depth discussion of previous studies comparing these methods in this population, if they exist, and from including more recent references (from within the last 5 years). The authors do not provide a reference from less than 5 years ago in the introduction.   Materials and Methods
This section needs to be completely rewritten. It would be helpful to provide the reader with a more appropriate structure for this section, including the type of study and its design, population and sample (inclusion and exclusion criteria); instruments and measures; procedures; ethics; and statistical analysis.
Lines 87-90 ask the authors to explain what they mean by this: The research material consists of studies conducted in the years 1999-2005 and 2003-2008 as part of health programs implemented by the Center for the Promotion of Children's Health and Fitness in Gdansk. The size of the study group divided by gender is shown in Table 1. Do you mean that you conducted a systematic review? A longitudinal study? This reviewer believes this paragraph is inadequate.
Line 91: This table is a description of the sample and therefore represents the results. This should be moved to the corresponding section.

More information is requested on how the validity and reliability of the height and weight measurements were ensured (how, by whom, and with what instruments). Describe the procedure if performed by more than one evaluator.
Regarding the statistical analysis, this evaluator should state that it is not considered correct since a three-way ANOVA (time x sex x age) should be performed, using time as a repeated measures factor [i.e., time (pre-test vs. post-test), sex (boys vs. girls), and age (6, 7, 8, 9, 10 years)] to analyze the possible main effect of these factors on the BMI variables, and their interaction using the Bonferroni statistic; the effect size was calculated in terms of eta squared (η2). Or if the authors decide to analyze the data by Ranges, they should perform a nonparametric test that takes into account the aforementioned factors.   Results
The results should be presented based on the requested reanalysis. They should begin with the classification of the sample by gender and age (sociodemographic data). This reviewer wonders how they can calculate the significance of the differences in the results. What statistical test was performed? Can the authors clarify this?
The tables should be improved by including footnotes indicating the meaning of some abbreviations (Tables 2, 3, 4, and 5).
Why are the results in Tables 7 and 8 in red?
They are presented clearly, with tables summarizing the quantitative data. However, they should be adapted to the reanalysis of the results requested from the authors.
The P value is not indicated for any of the comparisons.

When performing the reanalyses, the authors are suggested to provide a more detailed interpretation of the results, especially regarding the non-significant differences found. It is suggested that they include a more in-depth analysis of the possible reasons behind the non-significant differences and how these might influence the overall interpretation of the results.   Discussion The discussion should address the findings based on the requested new analyses and should also compare them with a similar study population with more recent references.

A section on the study's limitations should be included.
  Conclusion The conclusion should be based on the findings of the requested reanalysis of data and address the study's objective and/or question. It should also provide more detailed recommendations for future research, including suggestions on how to address the limitations identified in this study.   References The references are well selected, but none of the 32 are less than five years old. A high percentage of them should be less than five years old.
Please review the references to ensure that all citations are necessary and relevant to improve the quality of the manuscript.

Author Response

Dear Reviewer,
Thank you for taking the time to review and providing your feedback. It was very valuable to us.
We are sending the corrected text.

Round 2

Reviewer 1 Report

Comments and Suggestions for Authors

I have no further comments

Author Response

Dear Reviewer,

Thank you for reviewing our work again.

Best regards from the team

Reviewer 2 Report

Comments and Suggestions for Authors

The manuscript is suitable for pubblication

Author Response

(The authors gave the same response as above.)

Reviewer 3 Report

Comments and Suggestions for Authors

Dear authors,

I have thoroughly reviewed your manuscript again; unfortunately, you have not made any of the changes I suggested:
For example, I suggested that the title include the location where the study was conducted, the abstract include the values, or that the keywords could not be included in the title and therefore should be changed.
You have not made any of the changes, nor have I received a letter in response to my suggestions, either accepting or rejecting them. (This reviewer, like others, invests a lot of time trying to provide their perspective on improving the manuscript.) Furthermore, I expected a letter explaining the changes made, and if not, I expected a convincing explanation.

this manuscript presents serious methodological problems, as did all those I included in my previous review.

best regards

Author Response

Dear Reviewer,

We apologize for our oversight and lack of proper response. We come with answers.

Regarding the following comments:

(1)Title Authors are encouraged to indicate the location where the study was conducted to give the reader an idea of the context in which the research was conducted. - ADDED
(2)Abstract Authors are encouraged to indicate the p values for all comparisons, both significant and non-significant. - ADDED

(3)Keywords Keywords should not be included in the title. Authors are requested to include keywords not included in the title, such as BMI or longitudinal study. - CORRECTED

(4)Introduction (...)However, it could benefit from a more in-depth discussion of previous studies comparing these methods in this population, if they exist, and from including more recent references (from within the last 5 years). - ADDED

(5)Materials and Methods This section needs to be completely rewritten. It would be helpful to provide the reader with a more appropriate structure for this section, including the type of study and its design, population and sample (inclusion and exclusion criteria); instruments and measures; procedures; ethics; and statistical analysis. - CORREDTED

(6)recomendation to use ANOVA test - 

Thank you for this valuable suggestion. After taking a second look at the data, analyzing the purpose of the study and consulting with a professional statistician, we have confirmed that the method used — involving rank transformation and analysis of directional changes in BMI classification — is appropriate for the nature of our data. Given that BMI categories are ordinal and that our primary aim was to examine the direction and frequency of classification shifts over time, we opted for a descriptive and rank-based method rather than inferential modeling.

Moreover, due to the structure of our dataset — which includes categorical BMI classifications at two time points rather than continuous repeated measures — the application of repeated-measures ANOVA or its nonparametric alternatives would not accurately capture the primary dynamics we aimed to study.

We have therefore retained the original statistical approach.

(7) Limitation of the study - ADDED

(8) References - ADDED

The material was also checked by a language proofreader to catch linguistic errors.

We remain at your disposal.

Kind regards

Authors

Round 3

Reviewer 3 Report

Comments and Suggestions for Authors

.